# MicroRNAs in the Pathogenesis of Preeclampsia—A Case-Control In Silico Analysis

**Ramanathan Kasimanickam** [1,*] and **Vanmathy Kasimanickam** [2]

1   Department of Veterinary Clinical Sciences, College of Veterinary Medicine, Washington State University, Pullman, WA 99164, USA
2   Center for Reproductive Biology, College of Veterinary Medicine, Washington State University, Pullman, WA 99164, USA; vkasiman@wsu.edu
*   Correspondence: ramkasi@wsu.edu; Tel.: +1-509-335-6060

**Abstract:** Preeclampsia (PE) occurs in 5% to 7% of all pregnancies, and the PE that results from abnormal placentation acts as a primary cause of maternal and neonatal morbidity and mortality. The objective of this secondary analysis was to elucidate the pathogenesis of PE by probing protein–protein interactions from in silico analysis of transcriptomes between PE and normal placenta from Gene Expression Omnibus (GSE149812). The pathogenesis of PE is apparently determined by associations of miRNA molecules and their target genes and the degree of changes in their expressions with irregularities in the functions of hemostasis, vascular systems, and inflammatory processes at the fetal–maternal interface. These irregularities ultimately lead to impaired placental growth and hypoxic injuries, generally manifesting as placental insufficiency. These differentially expressed miRNAs or genes in placental tissue and/or in blood can serve as novel diagnostic and therapeutic biomarkers.

**Keywords:** preeclampsia; pathogenesis; transcriptome; miRNA; bioinformatics

## 1. Introduction

Preeclampsia (PE) is a complication of pregnancy with symptoms of high blood pressure, proteinuria, or other signs of organ damage, and occurs in 5% to 7% of pregnancies. It is one of the leading causes of maternal morbidity. Annually, PE causes over 70,000 maternal deaths and 500,000 fetal deaths worldwide [1]. Risk factors for PE include first pregnancy; previous occurrence of PE; history of hypertension; chronic kidney disease; history of thrombophilia; pregnancy from in vitro fertilization; family history of PE; type 1 or type 2 diabetes; a body mass index (BMI) of $\geq 35$ kg/m$^2$; advanced maternal age ($\geq 40$ years); and prolonged interval since last pregnancy [2].

Genetic factors were associated with the occurrence of PE [3]. In a previous study by Moufarrej et al. (2022), marked cell-free RNA (cfRNA) transcriptomic changes were observed between normotensive and preeclamptic mothers early in gestation, well before the onset of PE symptoms [4]. Furthermore, their study validated a panel of 18 genes using cfRNA expression to identify the mothers at risk of preeclampsia at 5 to 16 weeks of gestation, long before the manifestation of clinical symptoms [4].

Preeclampsia that originates from abnormal placentation primarily causes maternal and neonatal morbidity and mortality [5,6]. However, the cause of the abnormal development of the placenta remains poorly understood [7,8]. Genes were found to be differentially expressed between PE and normal placenta tissues and were associated with PE pathogenesis [5]. Hence, studies have been focused on the genetic signature of the placenta from preeclampsia.

Recent advances in high-throughput in silico techniques portray experimental data into exemplified biological networks. Exploring these biological networks can disclose the role of individual proteins, protein–protein interactions (PPIs), and corresponding biological functions. This study intended to use the transcriptomic profiling of mRNA in preeclamptic (PE) and normal placentae from Gene Expression Omnibus (GSE149812) for

further in silico analysis to elucidate the involvement of placenta-specific miRNA in the pathogenesis of PE.

## 2. Materials and Methods

In this study, differentially expressed PE-associated genes were identified from transcriptome data of PE and normal placenta samples. The gene expression data (profiled by microarray) and clinical characteristics were downloaded from the Gene Expression Omnibus (GSE149812; https://www.ncbi.nlm.nih.gov/geo/query/acc.cgi?acc=GSE149812; Accessed 7 July 2023). In the primary data, the description included patients' clinical characteristics, tissue collection, RNA extraction, and microarray analysis methods. An excerpt of the description is provided below in Sections 2.1 and 2.2.

### 2.1. Patients and Tissue Collection

Placental biopsies were obtained during cesarean section from both normotensive patients ($n$ = 3) and those with preeclampsia ($n$ = 3) (early onset type of PE; <31 weeks of gestation). All patients involved in this study were recruited from the Department of Obstetrics and Gynecology, the Third Xiangya Hospital, Central South University, Hunan, China. Pieces of villous tissue ($0.5 \times 0.5 \times 0.5$ cm$^3$), approximately 2 cm beside the umbilical cord insertion, from the middle layer of the placenta midway between the maternal and fetal surfaces from different areas, were excised, excluding sites of hemorrhage, infarction, and fibrin deposition. Tissues were immediately placed in 1.0 mL RNAstore Reagent (CWbiotech Company, Taizhou, China), and then stored at $-80$ °C until use.

### 2.2. RNA Extraction and Microarray Analysis

Total RNA was extracted using TRIzol following the manufacturer's instructions. Cyanine-3 (Cy3) labeled complementary RNA (cRNA) was prepared from 0.5 µg RNA using the One-Color Low RNA Input Linear Amplification PLUS kit (Agilent Tech. Inc., Santa Clara, CA, USA), followed by RNAeasy column purification (Qiagen Inc., Valencia, CA, USA). The cRNA yield was checked by an ND-1000 Spectrophotometer. Then, 1.5 µg of Cy3-labeled cRNA (specific activity > 10.0 pmol Cy3/µg cRNA) was fragmented at 60 °C for 30 min in a reaction volume of 250 mL containing 1× Agilent fragmentation buffer and 2 × Agilent blocking buffer. On completion of the fragmentation reaction, 250 mL of 2 × Agilent hybridization buffer was added to the fragmentation mixture and hybridized to Phalanx Human OneArray ver. 6 Release 1 for 17 h at 65 °C in a rotating Agilent hybridization oven. After hybridization, microarray slides were washed for 1 min at room temperature with GE wash buffer 1 (Agilent) and 1 min with 37 °C GE wash buffer 2 (Agilent) and then dried immediately by brief centrifugation. Slides were scanned immediately after washing on an Agilent DNA Microarray Scanner (G2505B) using one color scan setting for 1 × 44k array slides (scan area of 61 × 21.6 mm$^2$; scan resolution of 10 µm; dye channel set to Green, and Green PMT was set to 100%). The scanned images were analyzed using Feature Extraction Software 9.1 (Agilent).

### 2.3. Data Processing

The data were analyzed with GEO2R to identify genes that are differentially expressed between the two groups. GEO2R uses DESeq2, which is an R package for identifying differentially expressed genes from RNA-seq data [9,10] using negative binomial generalized linear models, which are suitable for studies with few replicates [10]. A 5-fold relative difference ($p \leq 0.05$) was used as a cut-off for the selection of differentially expressed (upregulated and downregulated) genes for further in silico analysis.

### 2.4. In Silico Analysis

2.4.1. Prediction and Analysis of Differentially Expressed Genes

The updated miRNet (http://www.mirnet.ca/, accessed on 1 July 2023) platform was used [11] to perform interaction analysis, separately, for upregulated and downregulated

genes. The degree (defined by the number of connections a node has to other nodes) and betweenness (defined by the number of connections occurring upon a node) of miRNAs and genes in the network were determined.

### 2.4.2. Gene Ontology and Functional Annotation Analysis of Genes with the Highest Degree and Betweenness Centrality

The top 20 up- and downregulated genes with the highest degree and betweenness centrality were selected, and their tissue expression, associated interacting genes (up to 6 genes; http://stringdb.org/; accessed on 6 July 2023), and single-cell normalized expression (https://www.proteinatlas.org/; accessed on 6 July 2023) were investigated.

### 2.4.3. Gene Ontology Enrichment and KEGG Pathway Analysis

All differentially expressed genes from the network were retrieved to recognize PPIs. The PPI network was created using the Search Tool for the Retrieval of Interacting Genes/Proteins (STRING) online database (http://stringdb.org/; accessed on 1 July 2023) separately for upregulated and downregulated genes [12]. Gene Ontology (GO) functional annotation for biological processes and Kyoto Encyclopedia of Genes and Genomes (KEGG) pathway enrichment analysis were also performed. A $p$-value of <0.05 was regarded as statistically significant.

### 2.4.4. Identification and Analysis of Hub Gene

The PPI networks for upregulated and downregulated genes from the STRING database were exported to Cytoscape software (version 3.10) [13]. The hub genes were selected as the top 20 nodes of the PPI network using the Maximal Clique Centrality (MCC) method [14], which has a better performance on the precision of predicting top essential proteins. Further analysis was performed using ClueGO [15] to integrate GO terms as well as KEGG pathways and create a functionally nested or organized GO/pathway term (k-score = 3). This task compares one set of genes or two lists of genes and comprehensively visualizes functionally grouped terms [15].

### 2.4.5. Gene Ontology and Functional Annotation Analysis of Hub Genes

The hub genes and their roles, tissue expression, and protein–protein interactions (up to 6 closely related genes) for differentially expressed genes in women with PE from STRING (http://stringdb.org/; accessed on 4 July 2023) and human protein atlas (https://www.proteinatlas.org; accessed on 4 July 2023) were investigated. To substantiate their presence, tissue expression and organelle localization were presented.

### 2.4.6. Comparison of miRNAs of Different Types of Preeclampsia

For comparison of different types (early- vs. late-onset; mild vs. severe) of preeclampsia, we selected DE genes in early-onset severe preeclampsia, late-onset severe preeclampsia, and late-onset mild preeclampsia from RNA-seq on 65 high-quality placenta samples that included 33 from 30 PE patients and 32 from 30 control subjects reported by Ren et al., 2021 [16]. These DE gene sets representing different types of PE were subjected to gene-miRNA interaction analysis.

## 3. Results

The transcriptomic (mRNA) profiling between PE and normal placenta tissues from Gene Expression Omnibus (GSE149812) recognized 28,254 genes (Supplementary File S1). There were 79 and 60 up- and downregulated genes, respectively (Supplementary File S1). Of those differentially expressed genes, 52 and 42 up- and downregulated genes, respectively, were at a 5-fold difference ($p \leq 0.05$; Supplementary File S2). The gene–miRNA interaction network analysis revealed the involvement of 45 upregulated and 32 downregulated genes.

From the gene–miRNA interaction network analysis, the degree and betweenness for the 45 upregulated genes were calculated. The 45 upregulated genes interacted with 829 miRNAs and 33 transcription factors (Figure 1). The degree and betweenness ranged from 1 to 19 and 0 to 16,641.0, respectively, for the 829 interacting miRNAs. The degree and betweenness ranged from 1 to 169 and 0 to 62,836.6, respectively, for the 45 upregulated genes. The degree and betweenness of the gene–miRNA interaction network for upregulated miRNAs is shown in Supplementary File S2.

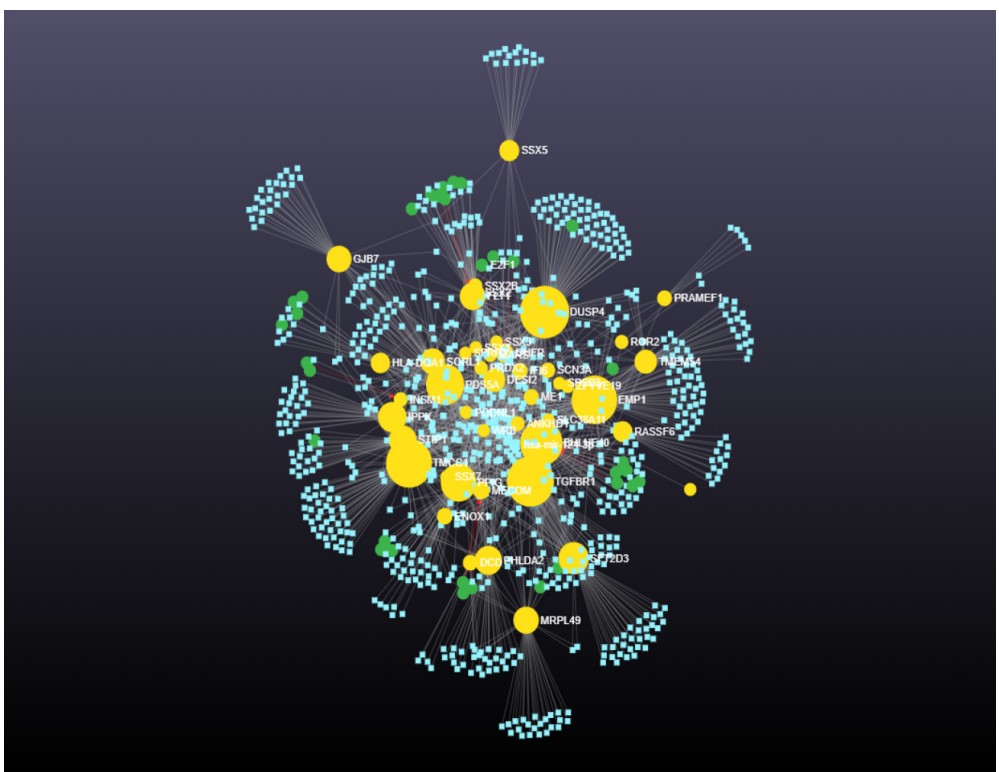

**Figure 1.** Gene–miRNA interaction networks for upregulated genes including 45 upregulated genes that interacted with 829 miRNAs and 33 transcription factors ($p < 0.05$). Green circles denote genes. Yellow circles denote transcription factors. Size indicates significance. Blue squares denote miRNAs.

Similarly, from the gene–miRNA interaction network analysis, the degree and betweenness for 36 downregulated genes were calculated. The 36 downregulated genes interacted with 1057 miRNAs and 39 transcription factors (Figure 2). The degree and betweenness ranged from 1 to 19 and 0 to 24,476.6 for the 1057 interacting miRNAs. The degree and betweenness ranged from 1 to 223 and 0 to 161,133.4 for the 36 downregulated genes. The degree and betweenness of the gene–miRNA interaction network for the downregulated miRNAs is shown in Supplementary File S3.

The interaction network for the top 20 upregulated genes is presented in Figure 3. The degree and betweenness ranged from 28 to 129 and 12,741.0 to 62,386.6 for the top 20 upregulated genes (Table 1). The interaction network for the top 20 downregulated genes is presented in Figure 4. The degree and betweenness ranged from 44 to 223 and 22,680.9 to 161,133.4 for the top 20 downregulated genes (Table 2). In addition, the top up- and downregulated genes' tissue expressions, single-cell normalized expressions (https://www.proteinatlas.org/; accessed on 7 July 2023), and functions are given in Tables 3 and 4, respectively.

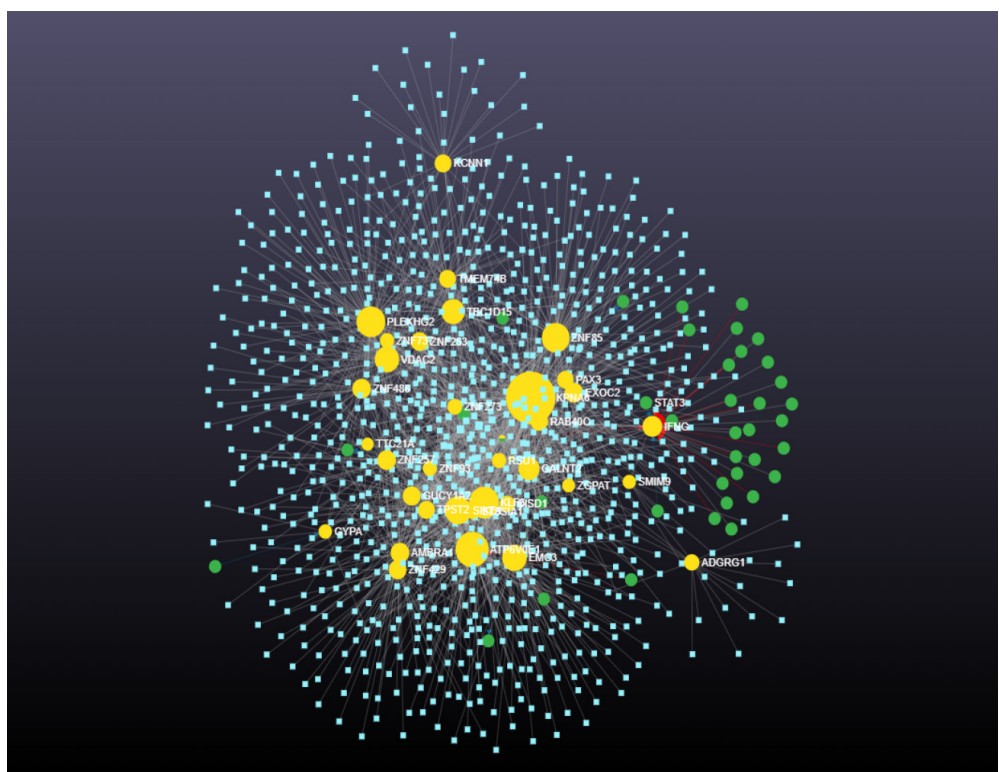

**Figure 2.** Gene–miRNA interaction network for downregulated genes including 36 downregulated genes that interacted with 1057 miRNAs and 39 transcription factors ($p < 0.05$). Green circles denote genes. Yellow circles denote transcription factors. Size indicates significance. Blue squares denote miRNAs.

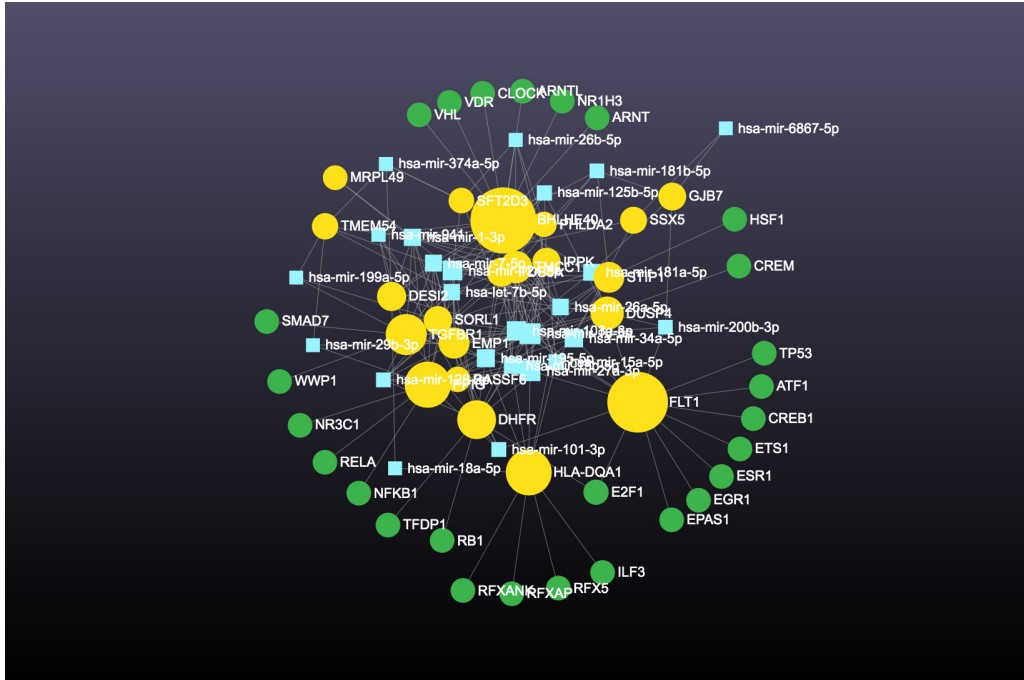

**Figure 3.** Gene–miRNA interaction network for the top 20 upregulated genes. Green circles denote genes. Yellow circles denote transcription factors. Size indicates significance. Blue squares denote miRNAs.

**Table 1.** Top 20 upregulated genes in the placenta with high degree and betweenness centrality in preeclamptic compared to normotensive women.

| High Degree Centrality | | | | High Betweenness Centrality | | | |
|---|---|---|---|---|---|---|---|
| # | ID | Degree | Betweenness | # | ID | Degree | Betweenness |
| 1 | TGFBR1 | 129 | 62,386.63603 | 1 | DUSP4 | 124 | 66,214.0901 |
| 2 | DUSP4 | 124 | 66,214.0901 | 2 | TGFBR1 | 129 | 62,386.63603 |
| 3 | TMCC1 | 122 | 60,207.54204 | 3 | TMCC1 | 122 | 60,207.54204 |
| 4 | EMP1 | 113 | 59,488.02209 | 4 | EMP1 | 113 | 59,488.02209 |
| 5 | BHLHE40 | 111 | 53,832.72771 | 5 | BHLHE40 | 111 | 53,832.72771 |
| 6 | PDS5A | 105 | 46,221.20935 | 6 | PDS5A | 105 | 46,221.20935 |
| 7 | PPIG | 96 | 41,670.73543 | 7 | PPIG | 96 | 41,670.73543 |
| 8 | IPPK | 70 | 28,805.24642 | 8 | SFT2D3 | 61 | 32,179.26096 |
| 9 | STIP1 | 65 | 27,238.03764 | 9 | IPPK | 70 | 28,805.24642 |
| 10 | DESI2 | 62 | 17,175.83835 | 10 | STIP1 | 65 | 27,238.03764 |
| 11 | SFT2D3 | 61 | 32,179.26096 | 11 | PHLDA2 | 52 | 26,712.22297 |
| 12 | SORL1 | 59 | 21,899.50057 | 12 | FLT1 | 57 | 23,913.42422 |
| 13 | FLT1 | 57 | 23,913.42422 | 13 | MRPL49 | 44 | 23,540.21993 |
| 14 | PHLDA2 | 52 | 26,712.22297 | 14 | GJB7 | 40 | 22,523.85633 |
| 15 | MRPL49 | 44 | 23,540.21993 | 15 | SORL1 | 59 | 21,899.50057 |
| 16 | GJB7 | 40 | 22,523.85633 | 16 | TMEM54 | 36 | 18,180.58635 |
| 17 | TMEM54 | 36 | 18,180.58635 | 17 | DESI2 | 62 | 17,175.83835 |
| 18 | **DHFR** | **34** | **10,104.03447** | 18 | **SSX5** | **22** | **13,882.13629** |
| 19 | RASSF6 | 32 | 13,330.8966 | 19 | RASSF6 | 32 | 13,330.8966 |
| 20 | HLA-DQA1 | 28 | 12,741.03278 | 20 | HLA-DQA1 | 28 | 12,741.03278 |

All genes that showed high degree centrality also had high betweenness centrality except the gene in bold letters.

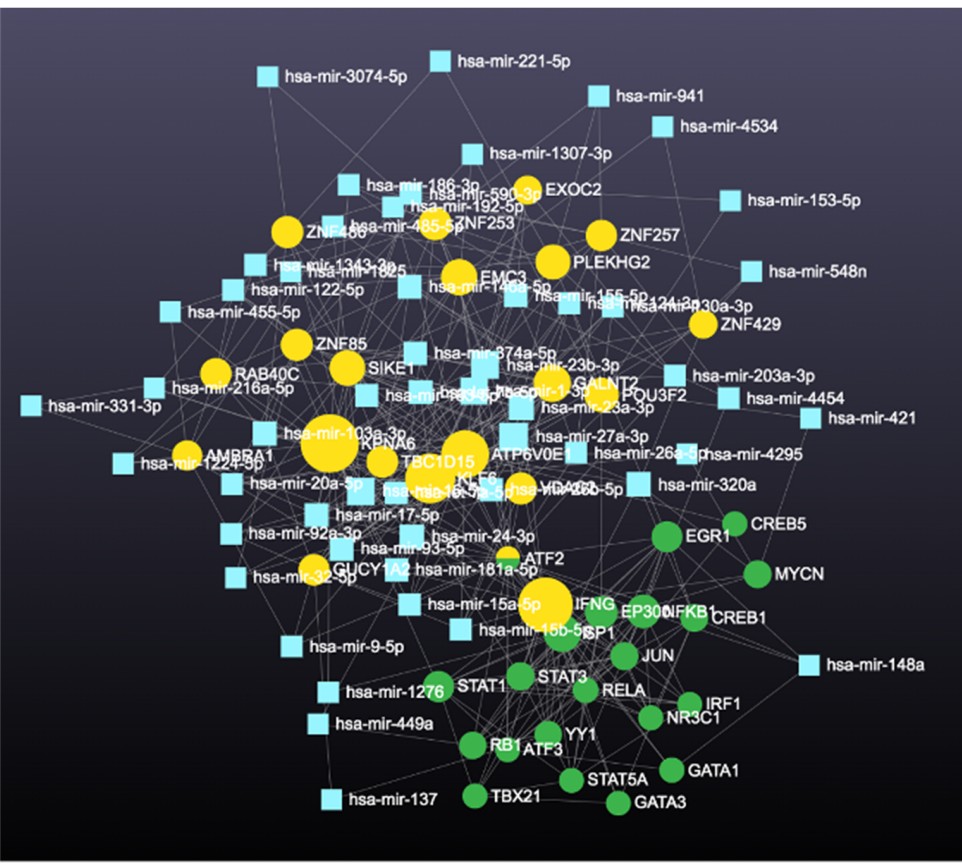

**Figure 4.** Gene–miRNA interaction network for the top 20 downregulated genes. Green circles denote genes. Yellow circles denote transcription factors. Size indicates significance. Blue squares denote miRNAs.

**Table 2.** Top 20 downregulated genes in the placenta with high degree and betweenness centrality in preeclamptic compared to normotensive women.

| | High Degree Centrality | | | | High Betweenness Centrality | | |
|---|---|---|---|---|---|---|---|
| # | ID | Degree | Betweenness | # | ID | Degree | Betweenness |
| 1 | KPNA6 | 223 | 161,133.371 | 1 | KPNA6 | 223 | 161,133.371 |
| 2 | ATP6V0E1 | 152 | 90,977.3419 | 2 | ATP6V0E1 | 152 | 90,977.34186 |
| 3 | KLF6 | 129 | 75,754.0689 | 3 | KLF6 | 129 | 75,754.06887 |
| 4 | SIKE1 | 118 | 60,992.2768 | 4 | PLEKHG2 | 112 | 71,728.72334 |
| 5 | PLEKHG2 | 112 | 71,728.7233 | 5 | ZNF85 | 98 | 67,140.10746 |
| 6 | ZNF85 | 98 | 67,140.1075 | 6 | SIKE1 | 118 | 60,992.27675 |
| 7 | EMC3 | 92 | 53,114.7673 | 7 | EMC3 | 92 | 53,114.76726 |
| 8 | GALNT2 | 83 | 38,016.4527 | 8 | VDAC2 | 69 | 52,426.38509 |
| 9 | TBC1D15 | 83 | 48,514.111 | 9 | TBC1D15 | 83 | 48,514.11101 |
| 10 | ATF2 | 81 | 35,905.8297 | 10 | GALNT2 | 83 | 38,016.45269 |
| 11 | VDAC2 | 69 | 52,426.3851 | 11 | ATF2 | 81 | 35,905.8297 |
| 12 | AMBRA1 | 55 | 27,918.5407 | **12** | **IFNG** | **41** | **32,732.16167** |
| 13 | RAB40C | 51 | 23,353.295 | 13 | AMBRA1 | 55 | 27,918.54066 |
| 14 | ZNF257 | 51 | 27,233.8438 | 14 | ZNF486 | 49 | 27,772.96946 |
| 15 | ZNF429 | 51 | 24,069.767 | 15 | EXOC2 | 49 | 27,644.24328 |
| 16 | EXOC2 | 49 | 27,644.2433 | 16 | ZNF257 | 51 | 27,233.84385 |
| 17 | ZNF486 | 49 | 27,772.9695 | 17 | GUCY1A2 | 47 | 25,416.96916 |
| 18 | ZNF253 | 47 | 23,149.4803 | 18 | ZNF429 | 51 | 24,069.76705 |
| 19 | GUCY1A2 | 47 | 25,416.9692 | 19 | RAB40C | 51 | 23,353.29504 |
| **20** | **POU3F2** | **44** | **22,680.8773** | 20 | ZNF253 | 47 | 23,149.48028 |

All genes that showed high degree centrality also had high betweenness centrality except the genes in bold letters.

**Table 3.** Top 20 upregulated genes (in the placenta with a high degree and betweenness centrality) and their tissue and single-cell expressions, associated genes, and functions.

| Gene | Tissue Expression | Single-Cell Normalized Expression (nTPM) | Associated Genes | Functions |
|---|---|---|---|---|
| TGFBR1 | Ovary, uterus placenta | Cyto 22.1; Syncytio: 18.4; extravillous: 7.3; Endometrium 21.2 | FKBP1A, TGFB1, TGFB3, TGFBR2, SMAD7 | Regulates cellular process: proliferation, maturation, differentiation, motility, and apoptosis |
| DUSP4 | Ovary, uterus placenta | Cyto 3.0; Syncytio: 24.9; extravillous: 48.8; Endometrium 13.7 | MAPK1, MAPK3, MAPK7, MAPK8, MAPK9 | Regulates cell proliferation and differentiation |
| TMCC1 | Ovary, uterus placenta | Cyto: 10.4; Syncytio: 27.3; extravillous: 0.6; Endometrium 14.2 | PLEC, RSP10, RSP10-NUDT3, RSP12, RSP18A, RSP19 | Regulates endosome fission; endosome membrane tubulation; and membrane fission |
| EMP1 | Ovary, uterus placenta | Cyto: 0.7; Syncytio: 0.7; extravillous: 0.6; Endometrium 161.6 | CCL4, LPAR6, LAPTM4B, PMP22, SMIM3 | Regulates cell proliferation and migration |
| BHLHE40 | Ovary, uterus placenta | Cyto: 31,8; Syncytio: 165.5; extravillous: 94.7; Endometrium 68.0 | BTRC, HDAC1, RXRA, TP53, SMAP2 | Regulates circadian rhythm and cell differentiation |
| PDS5A | Ovary, uterus placenta | Cyto: 32.6; Syncytio: 37.7; extravillous: 39.0; Endometrium 47.3 | RAD21, SMC1A, SMC3, STAG2, WAPAL | Regulates chromatid cohesion during mitosis |
| PPIG | Ovary, uterus placenta | Cyto: 186.3; Syncytio: 241.4; extravillous: 200.9; Endometrium 146.2 | BUD31, PCBP1, PRPF8, PRPF19, SNW1 | Regulates folding, transport, and assembly of proteins, and pre-mRNA splicing |
| IPPK | Ovary, uterus placenta | Cyto: 10.9; Syncytio: 23.7; extravillous: 14.8; Endometrium 4.6 | EPB41L4A, FRMD5, LPAR1, MPKAPK5, VRK1 | Regulates DNA repair, endocytosis, and mRNA export |
| STIP1 | Ovary, uterus placenta | Cyto: 127.7; Syncytio: 210.3; extravillous: 143.4; Endometrium 48.8 | HSP8, HSPA1A, HSP90AA1, HSP90AB1, PTGES3 | Regulates heat shock proteins |
| DESI2 | Ovary, uterus placenta | Cyto: 30.7; Syncytio: 43.8; extravillous: 42.9; Endometrium 39.9 | DDX5, E2F8, NPM1, NUP107, RPA1, UBE21 | Regulates protein deubiquitination |
| SFT2D3 | Ovary, uterus placenta | Cyto: 4.3; Syncytio: 3.0; extravillous: 4.2; Endometrium 8.9 | ADHFE1, ADACC, COG1, PSAT1, TMEM24, TSGA13 | Regulates protein transport and vesicle-mediated transport |
| SORL1 | Ovary, uterus placenta | Cyto: 0.2; Syncytio: 0.4; extravillous: 2444.5; Endometrium 2.9 | APP, APOE, CGA1, LRPAP1, VPS35 | Regulates protein transport |
| FLT1 | Ovary, uterus placenta | Cyto: 182.7; Syncytio: 10,058.3; extravillous: 980.8; Endometrium 1.4 | KDR, PGF, PTPN11, VEGFA, VEGFB | Regulates angiogenesis and vasculogenesis |
| PHLDA2 | Ovary, uterus placenta | Cyto: 4565.5; Syncytio: 365.0; extravillous: 336.1; Endometrium 27.9 | RANBP9, SUCO, SRC | Regulates fetal and placental growth |
| MRPL49 | Ovary, uterus placenta | Cyto: 63.8; Syncytio: 119.5; extravillous: 49.1; Endometrium 11.3 | COX15, TIMM10, METTL18, NXF1, FBXW11 | Regulates protein metabolism and mitochondrial translation |
| GJB7 | Ovary, uterus placenta | Cyto: 10.7; Syncytio: 8.9; extravillous: 4.7; Endometrium 0.7 | ARVCF, FYN, PAG1, PPP2R5E, ULBP2 | Regulates gap junction trafficking and vesicle-mediated transport |
| TMEM54 | Ovary, uterus placenta | Cyto: 48.2; Syncytio: 64.7; extravillous: 169.6; Endometrium 16.9 | CREB3, CDK2, HDAC1, LMNA, PEX19, RARA | Regulates membrane function |
| DHFR | Ovary, uterus placenta | Cyto: 34.5; Syncytio: 12.1; extravillous: 40.1; Endometrium 6.9 | FOX1, HSPD1, MDM2, FKBP1A, TP53, | Regulates folate metabolism and glycine and purine synthesis |
| RASSF6 | Ovary, uterus placenta | Cyto: 54.5; Syncytio: 48.9; extravillous: 24.8; Endometrium 2.0 | AMY1A, DLG1, KDM3A, HECTD1, SAV1, STK4 | Regulates cell cycle arrest and apoptosis |
| HLA-DQA1 | Ovary, uterus placenta | Cyto: 6.9; Syncytio: 4.8; extravillous: 10.7; Endometrium 33.4 | CD74, HLA-DQB1, KCNJ8, ST7, SLC38A9, TMEM214 | Regulates immune function |
| SSX5 | Ovary, uterus placenta | Cyto: 0; Syncytio: 0; extravillous: 0; Endometrium 0 | AGTRAP, PCBD2, NFE2, SSX2, ZSCAN1 | Regulates immune function |

Cyto—Cytotrophoblast; Syncytio—syncytiotrophoblast; extravillous—extravillous trophoblast; Endometrium—endometrial stromal cells.

**Table 4.** Top 20 downregulated genes (in the placenta with a high degree and betweenness centrality) and their tissue and single-cell expressions, associated genes, and functions.

| Gene | Tissue Expression | Single-Cell Normalized Expression (nTPM) | Associated Genes | Functions |
|---|---|---|---|---|
| KPNA6 | Ovary, uterus placenta | Cyto 41.2; Syncytio: 138.3; extravillous: 37.6; Endometrium 39.7 | HDAC1, KPNB1, LMNA, NUP50, RELB | Regulates protein transport |
| ATP6V0E1 | Ovary, uterus placenta | Cyto 511.0; Syncytio: 985.2; extravillous: 643.8; Endometrium 199.9 | ACP2, SLC7A2, CCDC115, PTPRF, TMEM199 | Regulates protein transport and pH of intercellular compartments |
| KLF6 | Ovary, uterus placenta | Cyto: 176.5; Syncytio: 217.0; extravillous: 539.4; Endometrium 616.8 | HDAC3, KLF4, LCOR, RELA, SP1 | Regulates cell growth |
| SIKE1 | Ovary, uterus placenta | Cyto: 30.8; Syncytio: 38.0; extravillous: 36.2; Endometrium 34.4 | PPP2R1A, PPP2CA, STRN4, STK24, STK25, TRAF3IP3 | Plays inhibitory role in virus- and TLR3-triggered IRF3 |
| PLEKHG2 | Ovary, uterus placenta | Cyto: 0.7; Syncytio: 0.6; extravillous: 2.9; Endometrium 18.6 | CDC42, GNB1, GNG2, RAC1, RHOA | Regulates lymphocyte chemotaxis via Rac and Cdc42 activation and actin polymerization |
| ZNF85 | Ovary, uterus placenta | Cyto: 10.5; Syncytio: 6.1; extravillous: 15.4; Endometrium 4.0 | CEP76, TRIM28 | Regulates DNA templated transcription |
| EMC3 | Ovary, uterus placenta | Cyto: 50.9; Syncytio: 91.6; extravillous: 57.7; Endometrium 50.2 | EMC1, EMC2, EMC4, EMC6, MMGT1 | Regulates membrane insertase activity |
| GALNT2 | Ovary, uterus placenta | Cyto: 6.9; Syncytio: 14.1; extravillous: 141.2; Endometrium 13.3 | AP4M1, AP4S1, MMGT1, MRPS5, ZMPSTE24 | Regulates glycosylation of protein |
| TBC1D15 | Ovary, uterus placenta | Cyto: 20.5; Syncytio: 48.2; extravillous: 16.3; Endometrium 39.6 | CCDC121, CEP23, OPTN, TBC1D17, UBXN8 | Regulates GTPase activator activity and mitochondrial morphology |
| ATF2 | Ovary, uterus placenta | Cyto: 13.9; Syncytio: 6.0; extravillous: 13.5; Endometrium 28.1 | FOS, JUN, MAPK8, MAPK9, MAPK14 | Regulates transcription of various genes involved in apoptosis, cell growth, proliferation, inflammation, and DNA damage response |
| VDAC2 | Ovary, uterus placenta | Cyto: 334.2; Syncytio: 399.4; extravillous: 470.9; Endometrium 107.0 | COX4I1, NDUFS4, PHB, PHB2, VDAC2 | Regulates oxidative metabolism, ion transport, cell apoptosis |
| AMBRA1 | Ovary, uterus placenta | Cyto: 3.9; Syncytio: 7.3; extravillous: 2.0; Endometrium 4.8 | BECN1, CUL4A, DDA1, DDB1, TCEB2 | Regulates mitophagy, cell proliferation, cell cycle progression |
| RAB40C | Ovary, uterus placenta | Cyto: 26.0; Syncytio: 51.3; extravillous: 15.8; Endometrium 6.7 | CUX2, CUX2, ENSP00000447000, RAB40B, SARNP | Regulates protein metabolism and autophagy |
| ZNF257 | Ovary, uterus placenta | Cyto: 4.2; Syncytio: 3.0; extravillous: 5.5; Endometrium 1.4 | HIST1H3A, SSRP1, CTCF, GL13, ZNF 513, ZNF710, ZNF768 | Regulates DNA templated transcription, apoptosis, protein folding and assembly, and lipid binding |
| ZNF429 | Ovary, uterus placenta | Cyto: 14.7; Syncytio: 12.7; extravillous: 11.7; Endometrium 10.3 | CTCF, GL13, ZNF 513, ZNF710, ZNF768 | Regulates transcription by RNA polymerase II, apoptosis, protein folding and assembly, and lipid binding |
| EXOC2 | Ovary, uterus placenta | Cyto: 15.3.; Syncytio: 13.9; extravillous: 6.6; Endometrium 6.2 | EXOC3, EXOC4, EXOC5, EXOC6, EXOC7 | Regulates polarized targeting of exocytic vesicles to specific docking sites on the plasma membrane |
| ZNF486 | Ovary, uterus placenta | Cyto: 4.6; Syncytio: 1.8; extravillous: 15.7; Endometrium 6.5 | CTCF, GL13, ZNF 513, ZNF710, ZNF768 | Regulates DNA templated transcription, apoptosis, protein folding and assembly, and lipid binding |
| ZNF253 | Ovary, uterus placenta | Cyto: 5.5; Syncytio: 4.5; extravillous: 3.4; Endometrium 5.2 | AKR1B1, LDOC1, CTCF, ZNF 513, ZNF710 | Regulates DNA templated transcription, apoptosis, protein folding and assembly, and lipid binding |
| GUCY1A2 | Ovary, uterus placenta | Cyto: 0.1; Syncytio: 0.2; extravillous: 0.0; Endometrium 2.0 | GUCY1B3, DLG1, DLG2, DLG3, DLG4 | Regulates conversion of GTP to 3′,5′-cyclic GMP and pyrophosphate |
| POU3F2 | Ovary, uterus placenta | Cyto: 0.0; Syncytio: 0.0; extravillous: 0.1; Endometrium 0.1 | POU4F1, POU4F2, POU4F3, SOX10, TFCP2 | Regulates neuronal differentiation and activation of CRH regulated genes |
| IFNG | Ovary, uterus placenta | Cyto: 0.1; Syncytio: 0.1; extravillous: 0.1; Endometrium 0.9 | FOXP3, IFNGR1, IFNGR2, RUNX1, TRIM2 | Regulates cellular response to viral and microbial infections |

Cyto—Cytotrophoblast; Syncytio—syncytiotrophoblast; extravillous—extravillous trophoblast; Endometrium—endometrial stromal cells.

After determining the degree and betweenness, the up- and downregulated genes that were 5-fold different ($p < 0.05$) were submitted (http://stringdb.org/; accessed on 7 July 2023) to elucidate enrichment networks. Figure 5 shows the PPIs for the upregulated genes (78 nodes; 193 edges; PPI enrichment with $p < 1.0 \times 10^{-16}$), revealing 225 significantly enriched biological process GO terms (False Recovery Rate, $p \leq 0.05$) and 54 significant (False Recovery Rate, $p \leq 0.05$) KEGG enrichment pathways (Supplementary File S2). Figure 6 shows the PPIs for the downregulated genes (73 nodes and 293 edges, PPI enrichment $p$-value of $<1.0 \times 10^{-16}$), revealing 268 significantly enriched biological process GO terms (False Recovery Rate, $p \leq 0.05$) and 87 significant (False Recovery Rate, $p \leq 0.05$) KEGG enrichment pathways (Supplementary File S3). The PPI networks for the up- and downregulated genes were separately constructed using the STRING database and Cytoscape software (Version 3.9). The top-ranked 20 hub genes using the Maximal Clique Centrality (MCC) method for up- and downregulated genes were screened and are presented in Figures 7 and 8, respectively. To interpret functionally nested gene ontology and pathway annotation networks for up- and downregulated genes in the PE placenta, ClueGo nested network analysis was performed, and the results are pre-

sented in Figure 9A–C and Figure 10A–C, respectively. The enrichment path from the ClueGo nested network analysis is presented in Supplementary File S4 (False Recovery Rate, $p < 0.05$). Tables 5 and 6 show the hub genes and their roles, tissue expressions, and protein–protein interactions (up to six closely related genes) for up-and downregulated genes in the PE placenta.

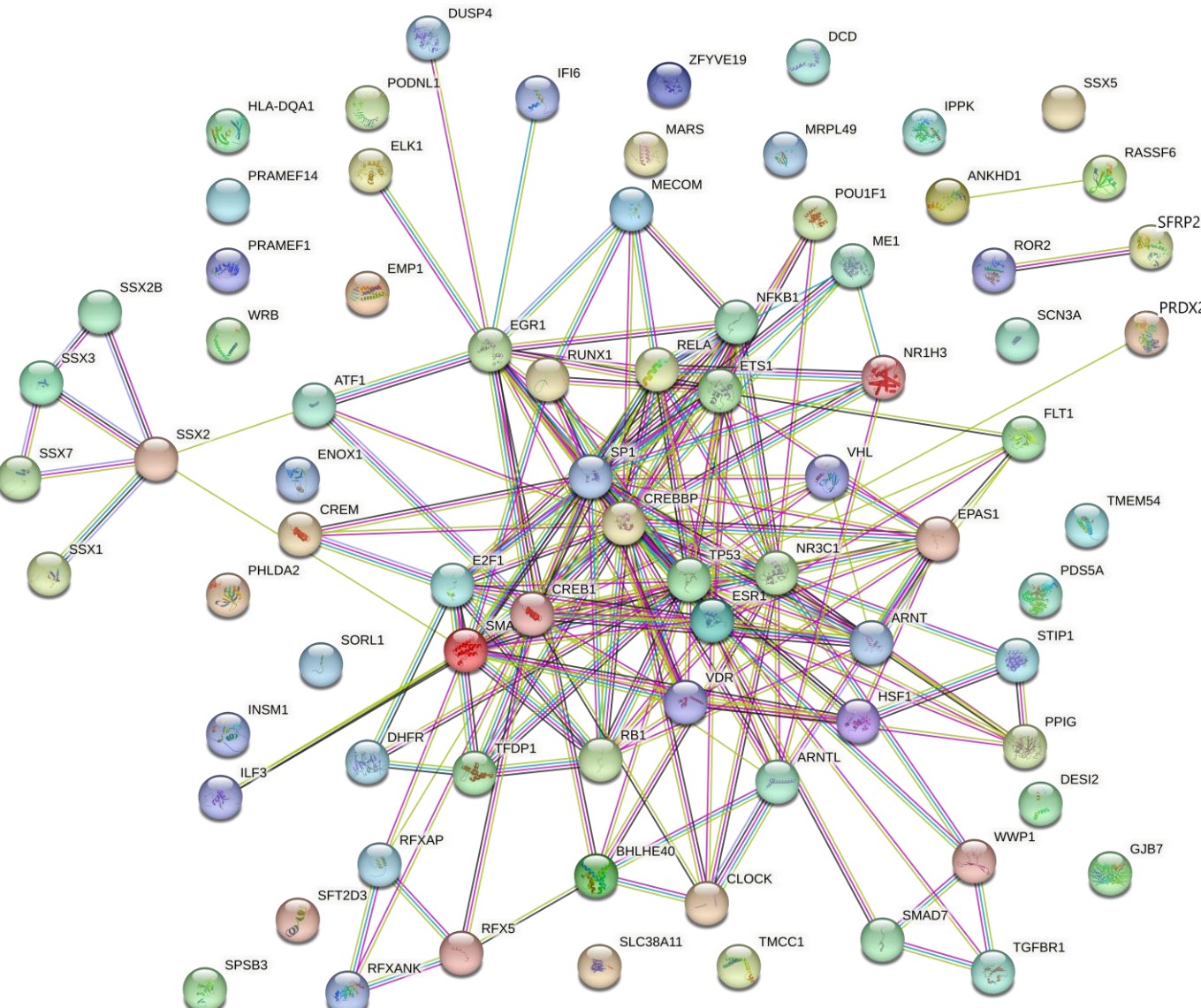

**Figure 5.** STRING protein–protein interaction (PPI) network. PPI network for the upregulated genes ($\geq$5-fold expression; 78 nodes; 193 edges; PPI enrichment with $p < 1.0 \times 10^{-16}$). The node color represents proteins. The edges represent interactions. Note: Some interacting proteins/transcription factors are common for upregulated and downregulated genes.

For the comparison of miRNAs of different types (early-onset severe preeclampsia, late-onset severe preeclampsia, and late-onset mild) of preeclampsia, the top 20 molecular markers (genes and miRNAs with high betweenness) were selected and compared. Six miRNAs (hsa-mir-124-3p, hsa-mir-1-3p, hsa-mir-146a-5p, hsa-mir-16-5p, hsa-mir-27a-3p, and hsa-mir-34a-5p) signifying all three PE types were recognized. Upon further comparison, it was realized that five (hsa-mir-1-3p, hsa-mir-146a-5p, hsa-mir-16-5p, hsa-mir-27a-3p, and hsa-mir-34a-5p) of these six miRNAs were the top miRNAs identified from the current analysis.

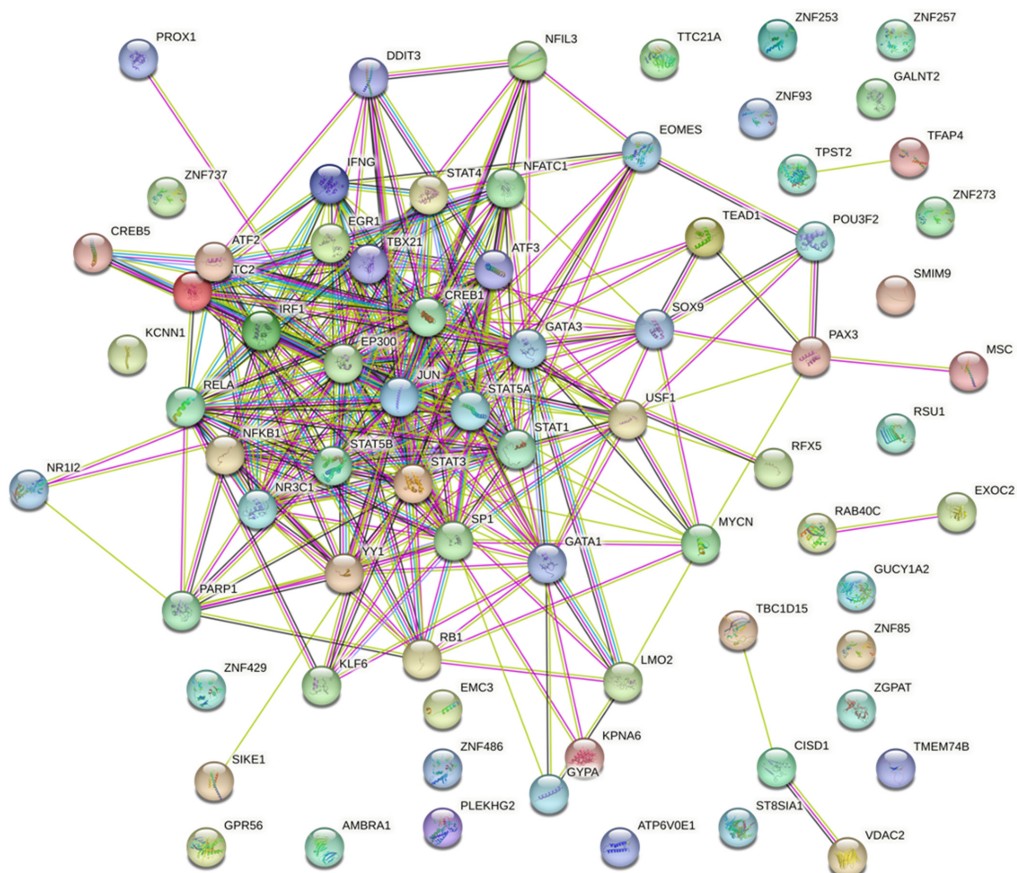

**Figure 6.** STRING protein–protein interaction (PPI) network. PPI network for the downregulated genes (≥5-fold expression; 73 nodes; 293 edges; PPI enrichment with $p < 1.0 \times 10^{-16}$). The node color represents proteins. The edges represent interactions. Note: Some interacting proteins/transcription factors are common for upregulated and downregulated genes.

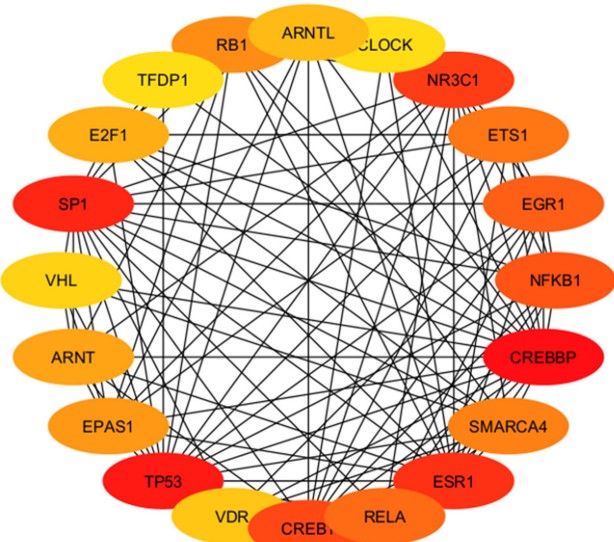

**Figure 7.** Interactions among hub genes (*ARNT, ARNTL, CLOCK, CREBBP, CREBP1, E2F1, EGR1, EPAS1, ESR1, ETS1, NFKB1, NR3C1, RB1, RELA, SMARCA4, SP1, TFD1, TP53, VDR* and *VHL*) of upregulated genes in the protein–protein interaction network. The dark to light colors denotes high to low degrees of expression. Black lines indicate interactions between genes.

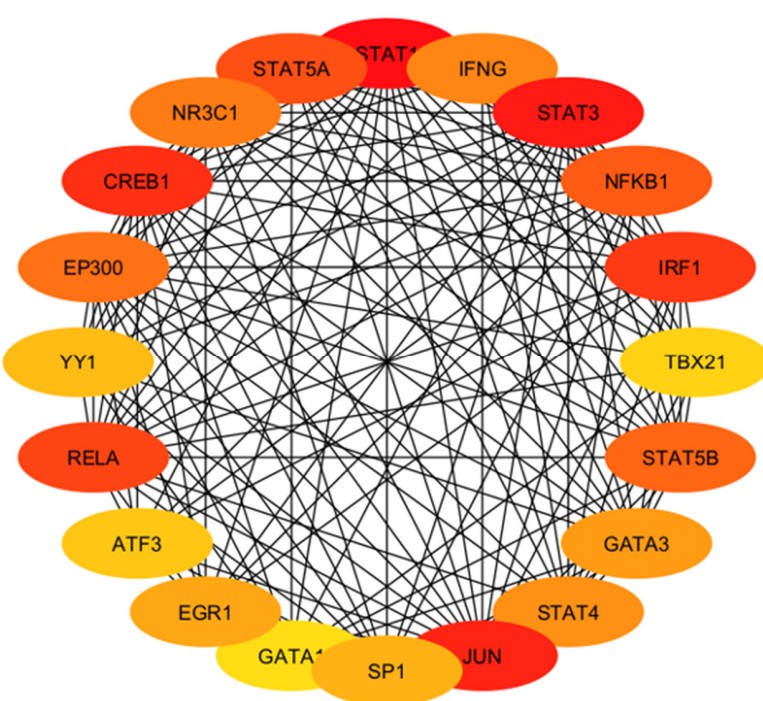

**Figure 8.** Interactions among hub genes (*ATF3*, *CREB1*, *EGR1*, *EP300*, *GATA1*, *GATA3*, *IFNG*, *IRF1*, *JUN*, *NFKB1*, *NR3C1*, *RELA*, *SP1*, *STAT1*, *STAT3*, *STAT4*, *STAT5A*, *STAT5B*, *TBX21*, and *YY1*) of downregulated genes in the protein–protein interaction network. The dark to light colors denotes high to low degrees of expression. Black lines indicate interactions between genes.

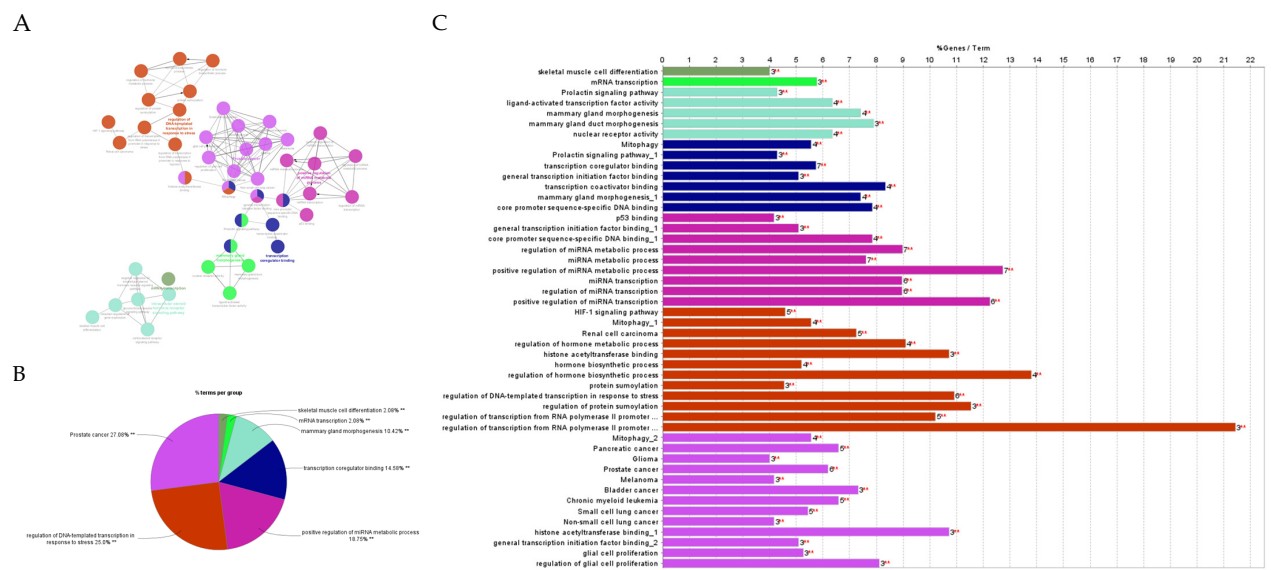

**Figure 9.** ClueGO analysis of upregulated genes. (**A**) Functionally grouped network with terms as nodes linked based on their kappa score level ($\geq$0.4), where only the label of the most significant term per group is shown. The node size represents the term enrichment significance. Functionally related groups partially overlap. The grey color gradient shows the gene proportion of each cluster associated with the term. (**B**) Overview chart with functional groups including specific terms for upregulated genes. ** $p < 0.001$. (**C**) GO/pathway terms specific for upregulated genes. The bars represent the number of genes (in red) associated with the terms. The percentage of genes per term is shown as a bar label.

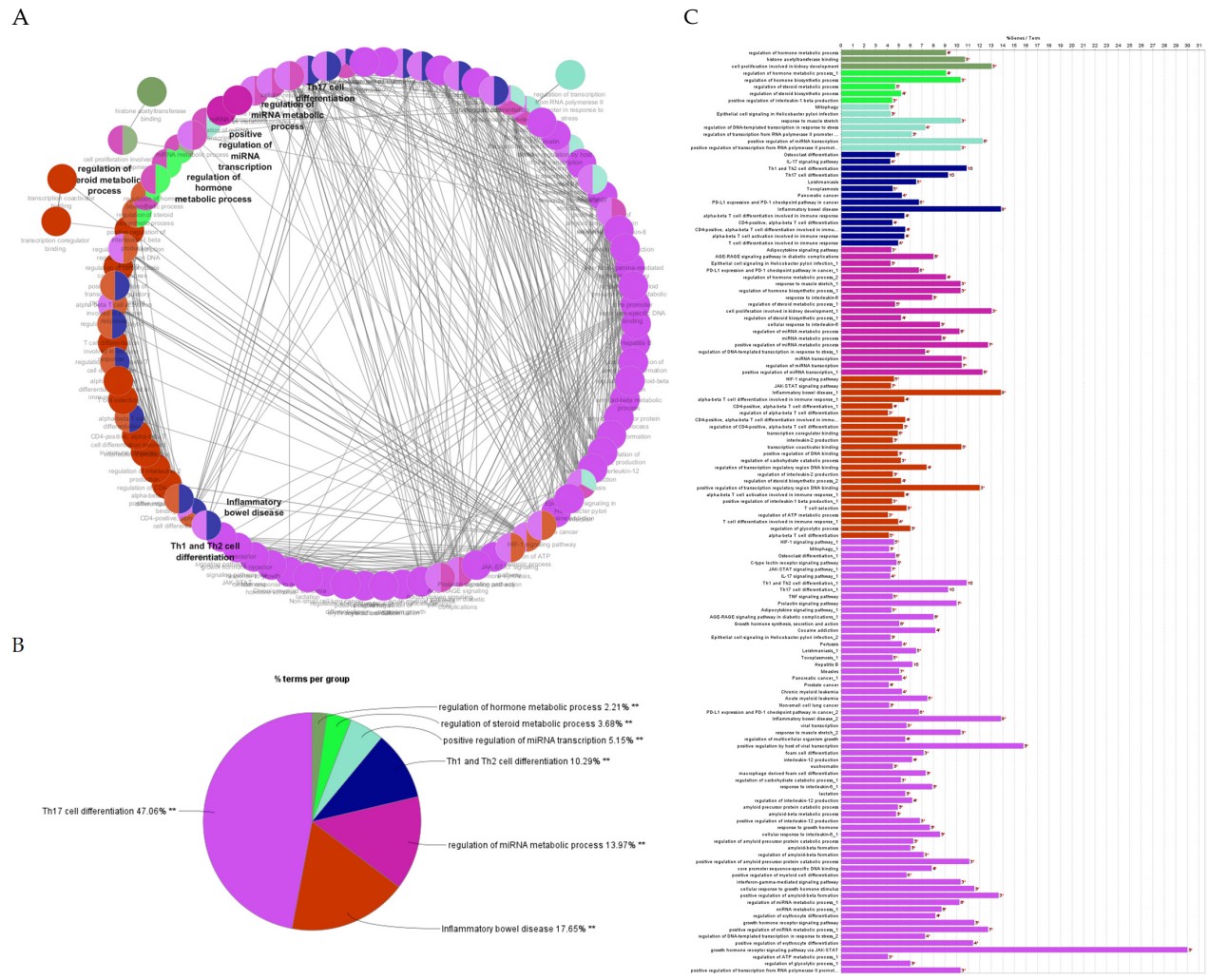

**Figure 10.** ClueGO analysis of downregulated genes. (**A**) Functionally grouped network with terms as nodes linked based on their kappa score level ($\geq$0.4), where only the label of the most significant term per group is shown. The node size represents the term enrichment significance. Functionally related groups partially overlap. The grey color gradient shows the gene proportion of each cluster associated with the term. (**B**) Overview chart with functional groups including specific terms for upregulated genes. ** $p < 0.001$. (**C**) GO/pathway terms specific for upregulated genes. The bars represent the number of genes associated with the terms. The percentage of genes per term is shown as a bar label.

**Table 5.** Top 20 upregulated hub genes and their tissue and single-cell expressions, associated genes, and functions.

| Hub Gene | Tissue Expression | Single-Cell Normalized Expression (nTPM) | Associated Genes | Functions |
|---|---|---|---|---|
| ARNTL | Ovary, uterus placenta | Cyto 17.0; Syncytio: 6.1; extravillous: 1.3; Endometrium 13.9 | CLOCK, CRY1 CRY2, NPAS2, PER2 | Regulates molecular circadian rhythm, myogenesis, adipogenesis, hormone production, cell proliferation |
| CLOCK | Ovary, uterus placenta | Cyto 11.3; Syncytio: 6.3; extravillous: 7.0; Endometrium 35.2 | ARNTL, CIPC, CRY1 CRY2, PER2 | Regulates molecular circadian rhythm |
| NR3C1 | Ovary, uterus placenta | Cyto: 48.6; Syncytio: 36.6; extravillous: 44.2; Endometrium 28.5 | HSP90AA1, NCOA1, NCOa2, NCOR, SMARCA4 | Regulates hypothalamic–pituitary–adrenal (HPA) axis by modulating availability of the cortisol |
| ETS1 | Ovary, uterus placenta | Cyto: 0.1; Syncytio: 0.3; extravillous: 0.4; Endometrium 49.7 | CREBBP, FOXO1, NFKB2, PAX5, RUNX1 | Regulates immune cell function |
| EGR1 | Ovary, uterus placenta | Cyto: 154.9; Syncytio: 165.7; extravillous: 106.1; Endometrium 783.3 | EP300, JUNDB, JUNDD, NAB1, TP53 | Regulates attachment and survival of normal cells and induces apoptosis in abnormal cells |

**Table 5.** *Cont.*

| Hub Gene | Tissue Expression | Single-Cell Normalized Expression (nTPM) | Associated Genes | Functions |
|---|---|---|---|---|
| NFKB1 | Ovary, uterus placenta | Cyto: 15.2; Syncytio:13.5; extravillous: 17.3; Endometrium 60.4 | NFKB1A, RELA, CHUK, IFBKB, RELB | Regulate genes |
| CREBBP | Ovary, uterus placenta | Cyto: 17.2; Syncytio: 32.2; extravillous: 11.1; Endometrium 52.1 | CREB1, HIF1A, KMT2A, MYB, TP53 | Regulates cell growth and division and prompting cells to mature and differentiate |
| SMARCA4 | Ovary, uterus placenta | Cyto: 72.5; Syncytio: 70.3; extravillous: 62.4; Endometrium 46.0 | SMARCB1, SMARCC1, SMARCC2, SMARCD1, SMARCE1 | Regulates chromatin remodeling |
| ESR1 | Ovary, uterus placenta | Cyto: 0.1; Syncytio: -; extravillous: -; Endometrium 72.4 | EP300, NCOA1, NCOA2, NR2F1, NR2F2 | Regulates many biological functions including growth, differentiation and function of female reproductive system, hormone binding, immune function |
| RELA | Ovary, uterus placenta | Cyto: 23.0; Syncytio: 47.7; extravillous: 27.7; Endometrium 24.8 | BRD4, CREBBPEP300, NFKB1, NFKB1A | Regulate genes involved in apoptosis, inflammation, the immune response, and proliferation |
| CREB1 | Ovary, uterus placenta | Cyto: 30.1; Syncytio: 18.7; extravillous: 25.9; Endometrium 37.8 | CREBBP, CRTC2, EP300, RPS6KA5, TP53 | Regulates proliferation, migration, and invasion of cells |
| VDR | Ovary, uterus placenta | Cyto: 0.1; Syncytio: 0.2; extravillous: 0.1; Endometrium 0.5 | NCOA1, NCOA2, NCOA3, MED1, RXRA | Induces a surge of cell signaling to maintain healthy $Ca^{2+}$ levels that serve to regulate several biological functions |
| TP53 | Ovary, uterus placenta | Cyto: 39.7; Syncytio: 20.4; extravillous: 40.6; Endometrium 28.3 | CREBBP, EP300, MDM2, MDM4, RPZ27A | Regulates cell division and apoptosis |
| EPAS1 | Ovary, uterus placenta | Cyto: 118.5; Syncytio: 365.0; extravillous: 336.1; Endometrium 31.3 | ARNT, EGLN1, VHL, TCEB1, TCEB2 | Regulates cell division, angiogenesis, adaptation to changing oxygen level |
| ARNT | Ovary, uterus placenta | Cyto: 24.1; Syncytio: 32.4; extravillous: 40.3; Endometrium 21.8 | AHR, EPAS1, HIF1A, NPAS3, SIM2 | Regulates placentation |
| VHL | Ovary, uterus placenta | Cyto: 35.3; Syncytio: 34.0; extravillous: 35.0; Endometrium 37.8 | EPAS1, CUL2, HIF1A, TCEB1, TCEB2 | Regulates cell growth and division |
| SP1 | Ovary, uterus placenta | Cyto: 16.3; Syncytio: 22.4; extravillous: 17.0; Endometrium 22.3 | EP300, ESR1, HDAC1, HDAC2, TP53 | Regulates cell cycle, hormonal activation, apoptosis, and angiogenesis |
| E2F1 | Ovary, uterus placenta | Cyto: 5.3; Syncytio: 2.1; extravillous: 8.8; Endometrium 1.0 | CCNA2, DP2, RB1, RBL1, TFDP1 | Regulates cell cycle progression, DNA repair, apoptosis |
| TFDP1 | Ovary, uterus placenta | Cyto: 85.2; Syncytio: 60.0; extravillous: 123.9; Endometrium 27.1 | E2F1, E2F4, E2F5, E2F6, RB1 | Regulates cell cycle progression |
| RB1 | Ovary, uterus placenta | Cyto: 6.9; Syncytio: 4.8; extravillous: 10.7; Endometrium 33.4 | CCND1, CDK4, DNMT1, E2F1, TFDP1 | Regulates cell growth and division |

Cyto—Cytotrophoblast; Syncytio—syncytiotrophoblast; extravillous—extravillous trophoblast; Endometrium—endometrial stromal cells.

**Table 6.** Top 20 downregulated hub genes and their tissue and single-cell expressions, associated genes, and functions.

| Hub Gene | Tissue Expression | Single-Cell Normalized Expression (nTPM) | Associated Genes | Functions |
|---|---|---|---|---|
| IFNG | Ovary, uterus placenta | Endometrium 0.9 | IFNGR1, IFNGR2, FOXP3, RUNX1, TRIM28 | Regulates cell differentiation, activation, expansion, homeostasis, and survival |
| STAT3 | Ovary, uterus placenta | Cyto 27.3; Syncytio: 35.9; extravillous: 49.3; Endometrium 194.6 | BMX, EGFR, JK1, MAPK1, PIAS3 | Controls cell proliferation, migration, apoptosis |
| NFKB1 | Ovary, uterus placenta | Cyto: 15.2; Syncytio:13.5; extravillous: 17.3; Endometrium 60.4 | NFKB1A, RELA, CHUK, IFBKB, RELB | Regulate genes |
| IRF1 | Ovary, uterus placenta | Cyto: 25.0; Syncytio: 9.4; extravillous: 46.9; Endometrium 179.7 | IRF8, STUB1, STAT1, EP300, KAT2B | Regulate innate and adaptive immune responses |
| TBX21 | Ovary, uterus placenta | - | CREBBP, EP300, GATA3, SP1, UBC, TBX21 | Regulates development of naive T lymphocytes |
| STAT5B | Ovary, uterus placenta | Cyto: 8.0; Syncytio: 13.8; extravillous: 5.9; Endometrium 20.5 | EGFR, INSR, JAK1, JAK2, JAK3 | Regulates formation of tissues and organs; maintains immune homeostasis |
| GATA3 | Ovary, uterus placenta | Cyto: 329.4; Syncytio: 1237.7; extravillous: 843.6; Endometrium 0.4 | HDAC1, HDAC2, HDAC3, LMO1, TAL1 | Regulates cell maturation with proliferation arrest and cell survival |
| STAT4 | Ovary, uterus placenta | Cyto: 0.4; Syncytio: 0.4; extravillous: 3.4; Endometrium 0.4 | JUN, IL12RB2, PIAS2, STAT1, ZNF467 | Regulates innate and adaptive immune responses |
| JUN | Ovary, uterus placenta | Cyto: 666.6; Syncytio: 405.9; extravillous: 61.9; Endometrium 2873.0 | ATF2, FOS, MAPK8, MAPK9, MAPK10 | Cell proliferation, apoptosis and survival, and tissue morphogenesis |
| SP1 | Ovary, uterus placenta | Cyto: 16.3; Syncytio: 22.4; extravillous: 17.0; Endometrium 22.3 | EP300, ESR1, HDAC1, HDAC2, TP53 | Regulates cell cycle, hormonal activation, apoptosis, and angiogenesis |
| GATA1 | Ovary, uterus placenta | - | BRD3, FLJI1, LMO2, TAL1, ZFPM1 | Regulates development of multipotential progenitors and hematopoietic stem cells |
| EGR1 | Ovary, uterus placenta | Cyto: 154.9; Syncytio: 165.7; extravillous: 106.1; Endometrium 783.3 | EP300, JUNDB, JUNDD, NAB1, TP53 | Regulates attachment and survival of normal cells and induces apoptosis in abnormal cells |
| ATF3 | Ovary, uterus placenta | Cyto: 179.2; Syncytio: 507.9; extravillous: 365.5; Endometrium 321.4 | DDIT3, JUN, JUNB, MDM2, TP53 | Regulates metabolism, immunity, inflammation, cell proliferation, and apoptosis |
| RELA | Ovary, uterus placenta | Cyto: 23.0; Syncytio: 47.7; extravillous: 27.7; Endometrium 24.8 | BRD4, CREBBPEP300, NFKB1, NFKB1A | Regulate genes involved in apoptosis, inflammation, the immune response, and proliferation |
| YY1 | Ovary, uterus placenta | Cyto: 121.3; Syncytio: 177.1; extravillous: 126.4; Endometrium 129.9 | EP300, HDAC2, HDAC3, MBTD1, RUVBL2, | Regulates several biological functions—embryogenesis, differentiation, replication, and cellular proliferation |

**Table 6.** *Cont.*

| Hub Gene | Tissue Expression | Single-Cell Normalized Expression (nTPM) | Associated Genes | Functions |
|---|---|---|---|---|
| EP300 | Ovary, uterus placenta | Cyto: 17.7; Syncytio: 34.4; extravillous: 19.0; Endometrium 49.1 | CITED2, HIF1A, SMAD3, TCF3, TP53 | Regulates cell growth and division and prompts cell maturation and cells to take specialized functions |
| CREB1 | Ovary, uterus placenta | Cyto: 30.1; Syncytio: 18.7; extravillous: 25.9; Endometrium 37.8 | CREBBP, CRTC2, EP300, RPS6KA5, TP53 | Regulates proliferation, migration, and invasion of cells |
| NR3C1 | Ovary, uterus placenta | Cyto: 48.6; Syncytio: 36.6; extravillous: 44.2; Endometrium 28.5 | HSP90AA1, NCOA1, NCOa2, NCOR, SMARCA4 | Regulates hypothalamic–pituitary–adrenal (HPA) axis by modulating availability of cortisol |
| STAT5A | Ovary, uterus placenta | Cyto: 1.2; Syncytio: 1.3; extravillous: 2.9; Endometrium 5.0 | EGFR, ERBB4, JAK1, JAK2, JAK3 | Relates IL2 signaling, modulates cytokine and growth factor action, modifies chromatin organization |
| STAT1 | Ovary, uterus placenta | Cyto: 13.7; Syncytio: 7.9; extravillous: 60.8; Endometrium 45.2 | CREBBP, JAK2, PIAS1, STAT2, STAT3 | Regulates proinflammation and immune function |

Cyto—Cytotrophoblast; Syncytio—syncytiotrophoblast; extravillous—extravillous trophoblast; Endometrium—endometrial stromal cells.

## 4. Discussion

Recent advances in high-throughput techniques transform experimental data into biological connotations. In illustrated networks, the nodes representing proteins, transcripts, or metabolites are linked by edges to show the interactions among nodes. Protein network exploration depicts the role of an individual protein and its communication with other proteins, representing the protein–protein interaction.

Centrality (network-based ranking of biological components) has been largely used to find important nodes in larger networks [17,18]. These nodes with higher degrees are more likely to be essential proteins influencing biological processes. These molecular markers and their properties are helpful when prioritizing them for disease associations. Using these methods, key biological mechanisms involved in the pathogenesis of PE were identified in the current study.

In this study, the gene–miRNA interaction networks of differentially expressed genes between PE and normal placentae revealed interactions with up to 28,000 genes and miRNAs. This shows the importance and depth of their involvement in the regulatory and interactive functions. Betweenness centrality measures the extent to which a miRNA/gene lies on paths between other miRNAs/genes. MicroRNAs/genes with high betweenness may have substantial influence within a regulatory network by virtue of their control over passing information between others [19]. It should be noted that genes with a high degree centrality are of important for the diagnosis of disease, and the proteins with a high degree of betweenness are important for drug discovery [20].

In this study, significantly upregulated (TGFBR1, DUSP4, TMCC1, EMP1, and BHLHE40) and downregulated (KPNA6, ATP6V0E1, KLF6, PLEKHG2, SIKE1, and ZNF85) genes with high degree and betweenness centrality showed key roles associated to the development of PE, including cell metabolic, developmental, proliferative, differentiative and apoptotic processes; cell macromolecule biosynthesis; DNA templated transcription; and responses to enzyme binding, stress, growth factor stimulation, lipid metabolism, and hypoxia.

### 4.1. Upregulated Genes with High Betweenness

Transforming growth factor beta 1 is a polypeptide member of the transforming growth factor beta superfamily of cytokines. It is a secreted protein that performs many cellular functions, including the control of cell growth, cell proliferation, cell differentiation, and apoptosis [21]. TGF-β1 signaling occurs by its binding with its receptor type 2 (TGFBR2), which in turn recruits and phosphorylates TGFBR1, forming a heterodimeric complex [22]. Once TGFBR1 is phosphorylated, it can downstream phosphorylate proteins SMAD2 and SMAD3, which then recruit SMAD4, translocate to the nucleus, and regulate the transcription of TGFβ1 target genes [23,24]. TGFβ1 levels were elevated in women with severe and mild preeclampsia late in gestation (mean gestational age, 40 weeks) compared with normotensive pregnant women [25–27]. TGFβ1 plays a decisive role in altering dNK (decidual natural killer) phenotype and function, which may have an obvious effect on the

pathogenesis of preeclampsia [20]. In the decidual zone of normal pregnancy, the dNK cell-mediated immune response and angiogenesis were subtly regulated by Treg cells via soluble TGFb1. However, in PE decidua, excessive amounts of TGFb produced by Treg cells could significantly impair the phenotype and function of dNK subpopulations. This distorted immune response may further damage decidual angiogenesis and cause pathological pregnancy [28]. In this investigation, TGFBR1 illustrated degree and betweenness scores of 129 and 62,386.6. The higher a gene's/protein's betweenness, the more important they are for the efficient flow of gains in a network, and downregulation of TGFBR1 would have had a significant impact on the biological functions and on the pathogenesis of PE.

The Dual-specificity phosphatase (DUSP) gene family is characterized by highly conserved amino acid sequences, implicated in a variety of biological functions [29]. Taurine upregulated 1 (TUG1) was downregulated in the placental tissues of PE patients compared with a control group [30]. TUG1 affected trophoblasts' biological function, including cell growth, migration, and crosstalk in vitro, and promoted the progression of preeclampsia. TET3 (tet methylcytosine dioxygenase 3, a DNA-binding protein) and DUSP were negatively regulated by TUG1. Molecular and functional interaction between TET3 and DUSPs impaired spiral artery remodeling in PE [30]. Downregulated TUG1 increased the expression of DUSP4 at both mRNA and protein levels. Notably, silencing of suppressor of variegation 39 homolog 1 (SUV39H1) by siRNAs significantly upregulated DUSP4, signifying the biding of TUG1 and SUV39H1 in the nucleus [30]. TET3 activated gene transcription by promoting DNA demethylation [31]. TET3 knockdown markedly decreased the cellular expression of DUSP4. In uterine cells, TET3 deficiency increased methylation of DUSP4 promoters. Further, the methylation level of DUSP4 promoters in the preeclamptic placenta was significantly increased compared with controls [30]. Overexpression of miR-218 (upregulated in this study; degree, 5 and betweenness, 609.1) significantly upregulated FOXP1 and TUG1 and downregulated DUSP4, at both mRNA and protein levels [30]. The regulatory network mediated by TUG1 and DUSP4 seems to be an essential determinant of the pathogenesis of PE, which regulates cell growth. In mice, the DUSP9 gene located on the X chromosome performs an essential function during placental development [31]. Mouse embryo lethality between 8 and 10.5 days postcoitum was due to a failure of labyrinth development. This correlates with the normal expression pattern of DUSP9 in the trophoblast giant cells and the labyrinth of the placenta.

Furthermore, TMCC1 was significantly downregulated in PE placentae compared with normal placentae [32]. EMP1 is a protein-coding gene involved in apoptosis, which negatively regulates cell growth [33]. Circulating EMP1 was positively associated with severe placental insufficiency, placental dysfunction, and fetal growth restriction [34]. BHLHE40 is a transcriptional repressor that responds to hypoxia and negatively regulates miR-196a-5p expression. BHLHE40/miR-196a-5p is involved in PE pathogenesis [35]. Knockdown of BHLHE40 or upregulation of miR-196a-5p restored cell viability, migration, invasion, and matrix metalloprotein (MMP)-2 and MMP-9 expression under hypoxia. BHLHE40 knockdown alleviated PE symptoms in pregnant C57/BL6N mice.

### 4.2. Downregulated Genes with High Betweenness Centrality

Karyopherin α6 (KPNA6, importin α7), directly interacts with the Kelch-like ECH Associated Protein 1 (KEAP1) [36]. Overexpression of KPNA6 facilitates KEAP1 nuclear import and attenuates the Nuclear Factor Erythroid 2-related Factor 2 (NRF2/NFE2L2) signaling, whereas knockdown of KPNA6 slows down KEAP1 nuclear import and enhances the NRF2-mediated adaptive response induced by oxidative stress [37]. Thus, KPNA6-mediated KEAP1 nuclear import plays an essential role in modulating the NRF2-dependent antioxidant response and maintaining cellular redox homeostasis [38]. In preeclampsia, there was increased decidual oxidative stress, NRF2-regulated gene expression was reduced, and KEAP1 protein expression was increased in areas of high trophoblast density [39]. This signifies the role of KPNA6. The degree and betweenness centrality scores for KPNA6 were 223 and 161,133.4. Regulatory networks mediated by KPNA6, KEAP1, and NRF2 are

essential determinants of the pathogenesis of PE, which regulates oxidative stress. ATPase H+ Transporting V0 Subunit E1 (ATP6V0E1/ATP6H) gene-regulated macro-autophagy was implicated in the pathogenesis of PE. ATP6H knockdown resulted in antiproliferative and apoptosis effects on BxPC-3 cells (pancreatic ductal adenocarcinoma cell line).

In normal pregnancies, placental autophagy is critical for the maintenance of cellular homeostasis that is needed for embryo and placental development [40]. Autophagy is activated in response to environmental stress, and dysregulation of autophagy is associated with various diseases [41]. Oxidative stress and hypoxia in preeclampsia are associated with an increase in the autophagic process, particularly in nutrient-deprived conditions [42]. Mitochondria are involved not only in ATP production but also in calcium homeostasis, free radical generation, cell survival, apoptosis, and necrosis [43–46]. Changes in mitochondrial dynamics, and apoptosis, are observed in preeclampsia [47]. Modification in mitochondrial gene expression influences mitochondrial homeostasis, ensuing mitochondrial dysfunction. This dysfunction leads to excessive ROS and inadequate ATP production [47,48]. Mitochondrial DNA (mtDNA) is speculated to be the marker of this dysfunction because of its inflammatory response. Oxidative stress causes membrane potential changes, inducing mitochondrial membrane depolarization and increased permeability. These disruptions will release damaged mitochondrial components, such as ROS and mtDNA, in the cytosol. As a result, there will be alteration in inflammatory and apoptotic pathways [49].

In PE, the mitochondrial apoptosis process seems to be highly altered [50]. There was a decrease in proapoptotic proteins such as p53 and BCL2-associated X and an increase in antiapoptotic proteins such as B-cell lymphoma 2 (BCL2) in term preeclamptic syncytiotrophoblast mitochondria compared with the increase in the BAX/BLC2 ratio in preterm preeclampsia [39]. In addition, soluble fms-like tyrosine kinase 1 (sFlt-1), which has antiangiogenic activity, exerted roles in oxidative stress and apoptotic pathways [51–53]. Differential apoptosis signaling in preterm and term placentae suggests that mitochondria promote cell survival in the placenta by suppressing the apoptosis mechanism. The regulation of programmed cell death and adequate antioxidant activity is important to improve mitochondrial adaptation and function [54]. Mitochondrial dysfunction due to excessive ROS production and reduced antioxidant capacity may result in an exaggerated apoptotic rate, placentation defect, and, therefore, preeclampsia.

The transcription factor Krüppel-Like Factor 6 (KLF6) has important roles in cell differentiation, angiogenesis, apoptosis, and proliferation. Furthermore, KLF6 is required for proper placental development [55]. KLF6 is present in both the early and late onset of severe-type PE [56]. KLF6 may mediate some of the effects of hypoxia in placental development and so has relevance in the development of PE.

PLEKHG2 is involved in cellular development, cellular assembly, and organization activity in early pregnancy and PE [57]. Decreased gene and protein expression of PLEKHG2 is involved in the breakdown of extracellular matrix proteins and tissue re-modeling activity in the human placenta [58]. Differentially expressed ZNF85 is involved in the top 10 GO terms, including DNA and ion bindings, between preeclampsia cases and controls [59]. In placental tissue, there was a correlation between ZNF85 expression and CpG methylation variation [60].

### 4.3. Comparison of miRNAs of Different Types of Preeclampsia

For comparison of different types (early- vs. late-onset; mild vs. severe) of preeclampsia, we selected DE genes in early-onset severe preeclampsia, late-onset severe preeclampsia, and late-onset mild preeclampsia from RNA-seq on 65 high-quality placenta samples that included 33 from 30 PE patients and 32 from 30 control subjects reported by Ren et al., 2021 [16]. These DE gene sets representing different types of PE were subjected to gene–miRNA interaction analysis. The top 20 molecular markers (genes and miRNAs with high betweenness) were compared, and the common six miRNAs (hsa-mir-124-3p, hsa-mir-1-3p, hsa-mir-146a-5p, hsa-mir-16-5p, hsa-mir-27a-3p, and hsa-mir-34a-5p) signifying all three types of PE types were identified. It is interesting to note that five (hsa-mir-1-3p, hsa-mir-146a-5p, hsa-mir-16-5p, hsa-mir-27a-3p, and hsa-mir-34a-5p) of these six miRNAs were the

top miRNAs (with high betweenness) exemplified from the current analysis. Their roles governing placenta development and PE are discussed below.

The significant alterations in the expression level of miRNA and the gene pairs hsa-miR-1-3p/ANXA2 and hsa-miR-1-3p/YWHAZ were associated with extracellular matrix organization, blood vessel development, smooth muscle contraction, angiogenesis, endothelial damage, and thrombi formation that caused a pulse increase in the right uterine and the umbilical arteries, hypoxia and oxidative stress, decreased placenta mass, and poor fetal development and weight (<10 percentile) [61].

Upregulated miR-146a-5p in the preeclamptic placentae provoked impaired trophoblast cell proliferation, poor invasiveness, and migratory capacity by inhibiting Wnt2 signaling [62]. miR-16-5p was upregulated in the placental tissue of a PE rat model [63]. miR-16-5p targeted the IGF-2 gene and downregulated its expression; consequently, it increased cell autophagy and cell death in the PE placenta [64].

In contrast, the downregulation of miR-27a-3p induced the migration and invasion of trophoblast cells into the uterine endometrium. Interestingly, the expression of miR-27a-3p was negatively related to ubiquitin-specific protease 25 (USP25) in recurrent miscarriage patients [65]. USP25 can regulate the processes of invasion and migration of different types of cells. It is reasonable that miR-27a-3p-mediated downregulation of USP25 contributes to the epithelial-to-mesenchymal transition, thereby inhibiting the migration and invasion of trophoblast cells via facilitating the Wnt pathway and regulating the miR-27a-3p/ATF3 axis [65,66].

miR-34a, a downstream gene of p53, regulates the cell cycle, apoptosis, and differentiation by targeting various target genes [67]. Elevated miR-34a has been reported to aggravate DNA damage and promote cell apoptosis [68]. Placental Growth Factor (PLGF) was a target gene of miR-34a [69]. PLGF regulates vascular endothelial growth and vascular remodeling via autocrine or paracrine mechanisms. miR-34a stimulates the proliferation of vascular endothelial cells and regulates DNA repair and apoptosis of these cells via PLGF [69]. It should be noted that hsa-miR-34a-5p was upregulated in the plasma during the first trimester in pregnant women with a high risk of preterm birth compared with normal controls. miR-34 was associated with pregnancy complications, including preeclampsia and intrauterine growth restriction [70,71].

### 4.4. Involvement of Hub Genes in Preeclampsia Development

ARNTL, CLOCK, NR3C1, ETS1, EGR1, NFKB1, CREBBP, SMARCA4, ESR1, RELA, CREB1, VDR, TP53, EPAS1, ARNT, VHL, SP1, E2F1, TFDP1, and RB1 proteins corresponding to the upregulated hub genes are involved in cellular proliferation, growth, and differentiation, cell metabolism, inflammation, and immune modulation in ovarian, uterine, and placental tissues (Table 5). In addition, biological rhythms and preeclampsia are linked [72], and ARNTL and CLOCK hub proteins are involved in circadian pathways. Similarly, IFNG, STAT3, NFKB1, IRF1, TBX21, STAT5B, GATA3, STAT4, JUN, SP1, GATA1, EGR1, ATF3, RELA, YY1, EP300, CREB1, NR3C1, STAT5A, and STAT1 proteins corresponding to downregulated hub genes are involved in cell survival, cellular growth and development, cell homeostasis, cell metabolism, immune modulation, and inflammation in ovarian, uterine, and placental tissues (Table 6). This demonstrates that the aberration of these hub genes results in PE instead of normal pregnancy.

### 4.5. Hub Genes with Diagnostic and Therapeutic Perspectives

Betweenness centrality measures the extent to which a miRNA/gene lies on paths between other miRNAs/genes. MicorRNAs/genes with high betweenness may influence information passing between others within the network [73]. The top three upregulated hub genes with high degree and betweenness scores were TGFBR1, DUSP4, and TMCC1. The top three downregulated hub genes with high degree and betweenness scores were KPNA6, ATP6V0E1, and KLF6. The higher a protein's betweenness, the more important it is for the efficient flow of goods in a network. It should be noted that proteins with a high

degree centrality are of important for the diagnosis of disease, and proteins with a high degree of betweenness are important for drug discovery [74].

Differentially methylated circadian clock genes ARNTL1, CLOCK, and BHLHE40 were observed in umbilical cord leukocytes and placental tissue in PE [75]. ARNTL and CLOCK are positive activators and drive the transcription of clock genes by binding to E-box elements on their promoters. The DNA methylation status of the circadian clock and clock-controlled genes in placental tissue and umbilical cord leukocytes is different between patients with EOPE and normal controls. This may be explained by a longer exposure to placental oxidative stress as compared with pregnancies complicated by late-onset preeclampsia. In term PE patients, the most enriched pathways that were correlated were hypoxia-related pathways and the membrane trafficking and autophagy-related pathways, which increased or decreased, respectively. Furthermore, CLOCK mRNA and protein expressions were reduced in the term PE placenta [76]. This suggests that circadian clock genes could be plausible candidates for the pathogenesis and etiology of PE.

The present work contains extensive bioinformatic analysis of genes, microRNAs, proteins, and biological processes between preeclampsia and normal pregnancy. However, this study may have limitations. The retrospective data extensively analyzed in this current study were originally obtained from relatively small biological samples. However, five-fold mean differences in relative expressions were used in this study. To detect these differences with adequate statistical power $(1 - \beta = 0.8)$ and statistical significance $(\alpha = 0.05)$, at least three samples per group were needed. The exact age (absolute age) of the pregnancy was not provided rather than stating that the normal and preeclampsia placental samples were obtained from less than 32 weeks of pregnancy. Early- and late-onset preeclampsia both result from the same problem, utero-placental malperfusion, which has different causes [77]. It has been suggested that early-onset preeclampsia is more strongly associated with internal placental factors, whereas the late-onset preeclampsia form may be primarily due to predisposing maternal factors. Some studies [78,79] found that the effect of risk factors varies according to the subtype of preeclampsia, whereas others did not [80]. Further, specific PE-related pregnancy complications are not distributed evenly across ages [81].

The association between the expression of placental tissue miRNAs and circulating miRNAs would help identify diagnostic and prognostic biomarkers. Cirkovic et al. (2021) observed increased miRNA-155 expression in both the placental tissue (SMD = 2.99, 95%CI = 0.83–5.14) and peripheral blood of women with PE (SMD = 2.06, 95%CI = 0.35–3.76) compared with women without PE [82]. However, an increased expression of miR-16a in placental tissue and significantly lower expression in peripheral blood of women with PE (SMD = –0.47, 95%CI = –0.91 to –0.03) was also observed. Several studies generated potential biomarkers utilizing samples from established PE, with less focus on prediction [83–87]. It is conceivable that coalescing biomarkers derived from different sources (multiple organ and cellular sources) may yield the best prediction. Utilizing large prospective cohort collections in unselected populations provides the best avenue for discovering novel biomarkers. However, miRNA expression differs according to the severity of PE [88] and during normal pregnancy [89]. So, these markers or combinations must be rigorously validated in external cohorts to ensure they achieve their potential to improve outcomes for pregnant people and their babies.

## 5. Conclusions

The evidence summarized in this article reveals the role of miRNAs in the pathogenesis of PE. The pathogenesis of PE is apparently determined by a range of miRNA molecules and their target genes and the degree of changes in their expression levels, which are associated with impairment of vascular and cellular development, circadian dysregulation, inflammation, and immunosuppression at the fetal–maternal interface, ultimately leading to impaired placental growth and hypoxic injury, which generally manifest as placental insufficiency. These miRNAs, genes, or proteins differentially expressed in placental tissue and in circulation can serve as novel diagnostic and therapeutic targets.

**Supplementary Materials:** The following supporting information can be downloaded at: https://www.mdpi.com/article/10.3390/cimb46040216/s1. Supplementary File S1: Transcriptomic profiling of mRNA (up- and downregulated at 5-fold relative expression) between PE and normal placentae. Supplementary File S2: STRING-based Gene Ontology terms for upregulated genes. Supplementary File S3: STRING-based Gene Ontology terms for downregulated genes. Supplementary File S4: Enrichment path from ClueGo nested network analysis.

**Author Contributions:** Conceptualization, V.K. and R.K.; methodology, V.K. and R.K.; software, V.K. and R.K.; validation, V.K. and R.K.; formal analysis, V.K. and R.K.; investigation, V.K. and R.K.; resources, V.K. and R.K.; data curation, V.K. and R.K.; writing—original draft preparation, R.K.; writing—review and editing, V.K. and R.K.; visualization, V.K. and R.K.; supervision, V.K. and R.K.; project administration, V.K. and R.K. All authors have read and agreed to the published version of this manuscript.

**Funding:** This research received no external funding.

**Institutional Review Board Statement:** Not applicable.

**Informed Consent Statement:** Not applicable.

**Data Availability Statement:** The data are available from the corresponding author upon reasonable request.

**Acknowledgments:** The authors thank the College of Veterinary Medicine, Washington State University, for the support.

**Conflicts of Interest:** The authors declare no conflicts of interest.

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
