# Peer review of "MicroRNAs in the Pathogenesis of Preeclampsia—A Case-Control In Silico Analysis"

_cimb, doi:10.3390/cimb46040216_

Round 1

Reviewer 1 Report

Comments and Suggestions for Authors

The study by Ramanathan Kasimanickam and colleagues aims to investigate miRNA in PE as compared to a healthy pregnancy. The scientific aim of the study is highly vivid as PE is a worldwide problem affecting both mother and fetal development. Thus research on pathogenesis that makes opportunities to identify future biomarkers is highly needed.

The pros of the given paper are the utilization of technical demanding molecular assay with advanced results analysis. In the introduction, the PE background and the scientific aims were properly formulated. The results were presented in an accessible way despite its technical complexity. The material and method section describes all the important steps of an applied assay that enables reproductivity. The results were described properly and the figures supplement the paper accordingly. Finally, the discussion section refers to the obtained results and puts them in the current state of knowledge.

Unfortunately, the major flaw of the given research is the highly limited sample number, that is only three samples of PE and three of control. The low sample number may affect results and change its interpretation. Some power analysis would help to judge the study outcome. Another is the lack of demographic and clinical characterization of participants. Having such a limited study group and control, detailed information on both PE and control individuals should be given. Another improvement could be a short discussion of how your results could be translated into real biomarkers identification in the clinics. How far, upon the data from the literature, your results can correspond with peripheral blood samples?

Author Response

We thank the reviewer for the comments and suggestions.

The power analysis has been included in the original version. Please refer lines 515 to 518 about sample size calculation. “The retrospective data extensively analyzed in this current study were originally obtained from relatively small biological samples. However, five-fold mean differences in relative expressions were used in this study. To detect these differences with adequate statistical power (1 − β = 0.8) and statistical significance (α = 0.05), at least three samples per group were needed.” Further, usually 2-fold relative differences were used as cut-off. We used 5-fold relative mean differences. 

Regarding the demographic and clinical characteristics, since this study is a secondary analysis, information available has been provided. Probable biomarker miRNAs have been selected and relevant statements have been provided.

To answer the last question, miRNAs in the blood are pretty stable but the levels are low (in other words, miRNAs are minimally expressed in the blood). Consequently, the correlation between circulatory and tissue miRNAs should be strong. However, we included a paragraph addressing the reviewer's suggestion at the end of discussion section.

Reviewer 2 Report

Comments and Suggestions for Authors

In the last 5 years, interest in the genetic study of preeclampsia has increased, with those related to miRNAs being more represented. These studies would help elucidate aspects of etiology, prediction1-3, diagnosis4 and treatment....

The article of Kasimanickam and Kasimanickam, it is an excellent work that tries to elucidate the pathogenesis of preeclampsia by analyzing the interaction between miRNA and its target genes.  The work must be published in its current format

1.            Yoffe L, Gilam A, Yaron O, et al. Early Detection of Preeclampsia Using Circulating Small non-coding RNA. Scientific reports. Feb 21 2018;8(1):3401. doi:10.1038/s41598-018-21604-6

2.            Zhou S, Li J, Yang W, et al. Noninvasive preeclampsia prediction using plasma cell-free RNA signatures. American journal of obstetrics and gynecology. Nov 2023;229(5):553.e1-553.e16. doi:10.1016/j.ajog.2023.05.015

3.            Ogoyama M, Takahashi H, Suzuki H, Ohkuchi A, Fujiwara H, Takizawa T. Non-Coding RNAs and Prediction of Preeclampsia in the First Trimester of Pregnancy. Cells. Aug 5 2022;11(15)doi:10.3390/cells11152428

4.            Morey R, Poling L, Srinivasan S, et al. Discovery and verification of extracellular microRNA biomarkers for diagnostic and prognostic assessment of preeclampsia at triage. Sci Adv. Dec 22 2023;9(51):eadg7545. doi:10.1126/sciadv.adg7545

Author Response

We thank the reviewer for the comments and suggestions.

These suggested publications relevant to circulating or extracellular miRNAs. Our study is the secondary analysis from the tissue miRNAs. However, we have added the suggested four references at the end of discussion section.

Reviewer 3 Report

Comments and Suggestions for Authors

This is a secondary analysis of the data published in GSE149812. Authors utilized this dataset and performed in-silico gene-miRNA interaction network analysis, differential expression, PPI analysis, and identified hub genes, pathways involved in PE placenta compared to the control that could be the potential therapeutic targets. Overall, the manuscript is well-written, and I have very little comments on the method and figures.

1)        Method 2.1 and 2.2: I assume this section is from the original paper that derived the dataset GSE149812. However, when I read this section, I thought these samples were involved in this paper, so I looked through the paper to figure out where these placental samples were used. I do not think you need these 2.1 and 2.2, because authors already stated in the paper that you are doing the secondary analysis of the GSE149812.

2)        Figure 9 and 10: I cannot read anything on this panel. Need to increase the font size.

Author Response

We thank the reviewer for the comments and suggestions.

A relevant statement has been added to the manuscript, stating that the methodology is from the primary experiment with the specific citation. We believe that inclusion of brief description in these sections 2.1 and 2.2 will avoid readers looking for the information.

We increase the font size images 9 and 10, but the images became too big and was unable to fit the page. So we did not modify the images. Experience with similar images from our previous publications, we believe that it will be resolved during the page formation.

Round 2

Reviewer 3 Report

Comments and Suggestions for Authors

I did not see any updates on my comments.

Author Response

We thank the reviewer for the comments and suggestions.

A relevant statement has been added to the manuscript, stating that the methodology is from the primary experiment with the specific citation (Refer to Lines 56 to 58). We believe that inclusion of data description in sections 2.1 and 2.2 will help readers looking for the information instead of referring to the original data. So we did not remove the sections 2.1 and 2.2.

We included all original figures of Fig. 9 & 10 in a separate image file for reference.

In the text when we increase the font size of figures 9 and 10, the images became too big and was unable to fit the page. So we did not modify the images in the text.

Round 3

Reviewer 3 Report

Comments and Suggestions for Authors

I do not have any comments to raise.